# Trends of *Mansonia* (Diptera, Culicidae, Mansoniini) in Porto Velho: Seasonal patterns and meteorological influences

**José Ferreira Saraiva** [1] *, **Nercy Virginia Rabelo Furtado**[1,2], **Ahana Maitra**[3], **Dario P. Carvalho**[4], **Allan Kardec Ribeiro Galardo**[1], **José Bento Pereira Lima**[5]

**1** Medical Entomology Laboratory, Instituto de Pesquisas Científicas e Tecnológicas do Estado do Amapá – IEPA, Macapá, Amapá, Brazil, **2** Postgraduate Program in Tropical Medicine, Instituto Oswaldo Cruz, Fiocruz, Manguinhos, Rio de Janeiro, Brazil, **3** Department of Pharmacy–Pharmaceutical Sciences, University of Bari "Aldo Moro", Bari, Italy, **4** Santo Antônio Energia–SAE, Porto Velho, Rondônia, Brazil, **5** Laboratory of Biology, Control, and Surveillance of Insect Vectors (LaBiCoVIV), Instituto Oswaldo Cruz (IOC), Fiocruz, Rio de Janeiro, Brazil

* jfsento@gmail.com

## Abstract

Entomological research is vital for shaping strategies to control mosquito vectors. Its significance also reaches into environmental management, aiming to prevent inconveniences caused by non-vector mosquitoes like the *Mansonia* Blanchard, 1901 mosquito. In this study, we carried out a five-year (2019–2023) monitoring of these mosquitoes at ten sites in Porto Velho, Rondônia, using SkeeterVac SV3100 automatic traps positioned between the two hydroelectric complexes on the Madeira River. Throughout this period, we sampled 153,125 mosquitoes, of which the *Mansonia* genus accounted for 54% of the total, indicating its prevalence in the region. ARIMA analysis revealed seasonal patterns of *Mansonia* spp., highlighting periods of peak density. Notably, a significant decreasing trend in local abundance was observed from July 2021 (25th epidemiological week) until the end of the study. Wind speed was observed to be the most relevant meteorological factor influencing the abundance of *Mansonia* spp. especially in the Joana D'Arc settlement, although additional investigation is needed to comprehensively analyze other local events and gain a deeper understanding of the ecological patterns of this genus in the Amazon region.

## Introduction

Entomological monitoring of mosquitoes represents an important part of public health data collection, as many of these insects are vectors of several human pathogens [1–3]. Information provided in the entomological survey includes species distribution, population density indices, and activity patterns. This data can help predict disease outbreaks and propose better vector control strategies [4]. Additionally, monitoring can reveal changes in insecticide resistance patterns, which is critical to the effectiveness of mosquito control interventions [3], making entomological monitoring an indispensable tool for the surveillance and control of diseases transmitted by mosquitoes [5].

**Data Availability Statement:** All relevant data are within the manuscript and its Supporting Information files.

**Funding:** Research and Development project from Santo Antônio Energia (ANEEL project CT. PD.124.2018).

**Competing interests:** The authors have declared that no competing interests exist.

The human landing catch (HLC), historically recommended for entomological monitoring studies of mosquitoes due to its efficiency in obtaining anthropophilic species [6, 7], presents risks of infection for collectors during catches [4]. On the other hand, light traps, such as CDC and Shannon, appear as an alternative for collection, reducing the dangers of infection [8]. However, these devices attract a wide variety of insects, sometimes collecting more insects from other groups than mosquitoes [9]. Recently, automatic traps utilizing pheromones have gained prominence, exemplified by the SkeeterVac SV3100 (Blue Rhino®, Winston Salem, NC), which employs synthetic pheromone refills based on Lurex and 1-octen-3-ol (Octenol) to mimic odors of mosquito hosts. Despite being widely used in homes to reduce mosquito attacks due to their high collection efficiency, these traps have been little explored as an alternative for the entomological monitoring of mosquitoes [10].

In Porto Velho, more precisely between the hydroelectric complexes of Santo Antônio Energia and Jirau, located on the Madeira River, studies on the malarial potential identified, from 2014 onwards, an increase in the density of *Mansonia* Blanchard, 1901. With its large size and notable aggressiveness in hematophagic activity, this mosquito has not yet been officially implicated as a disease vector in Brazil [11]. However, *Mansonia* has caused problems for human populations in the Amazon, where its presence is marked by high density, especially in the vicinity of muddy water backwaters and densely covered by aquatic macrophytes [12]. Its proliferation appears to correlate to environmental impacts and the introduction of large animals, such as cattle, used as a blood meal source [13].

In this study, we conducted long-term monitoring of *Mansonia* spp. using automatic traps, specifically the SkeeterVac SV3100 model. These traps were baited with two pheromones (Lurex and Octenol) to estimate the present and future abundance of these mosquitoes. Simultaneously, we sought to evaluate seasonality and its correlation with meteorological factors. The central objective is to provide a more in-depth understanding of the population dynamics of these mosquitoes, contributing to improving monitoring and control strategies.

## Material and methods

### Ethics statement

The present study was performed in accordance with scientific license number 65279–1 provided by SISBIO/IBAMA (Authorization and Information System on Biodiversity/Brazilian Institute of the Environment and Renewable Natural Resources) for capture of culicids in Brazilian national territory.

### Area of the study

The city of Porto Velho, the capital of the State of Rondônia, located in the Western Brazilian Amazon, covers a territorial extension of 35,000 square kilometers and has a population of 460,434 inhabitants (2019 census). These numbers represent less than 1% of the area and 2% of the total population of the Amazon region of Brazil [14].

*Mansonia* monitoring was conducted in the municipality of Porto Velho, specifically in areas adjacent to the capital. The locations chosen for the study are characterized as rural, semi-rural, or peri-urban environments, all close to the reservoirs of the Santo Antônio (SAE) and Jirau hydroelectric plants, as indicated in Fig 1. In this perimeter, where the hydroelectric complex of the Madeira river is also located, there is an Intense agricultural expansion of monocultures, such as soybeans, corn, and rice, in addition to extensive livestock farming. These activities have also driven logging, promoting deforestation in this region [15, 16].

According to the Köppen climate classification, the region has an Aw–rainy type climate, characterized by average temperatures that vary between 21˚C and 34˚C, with rare

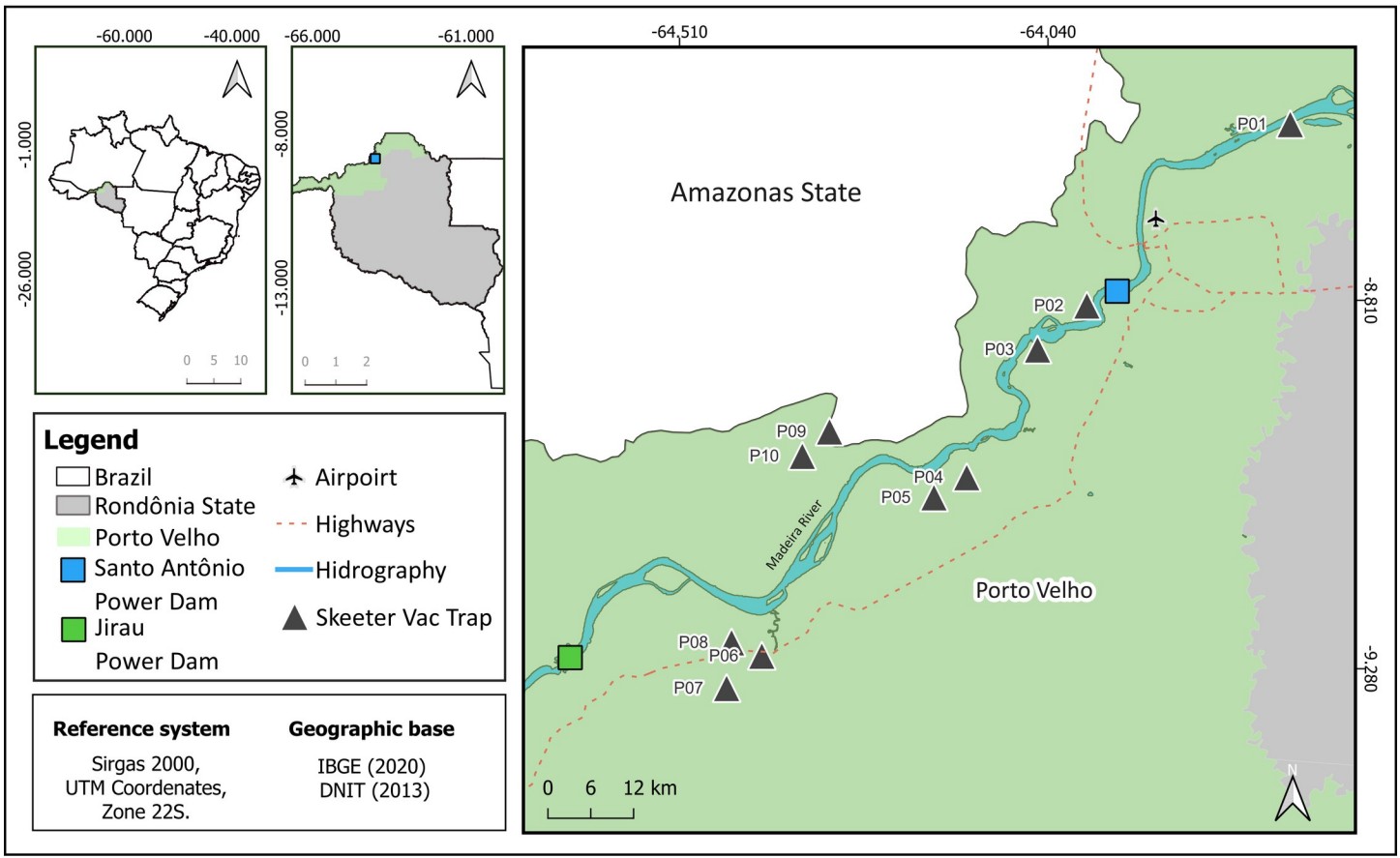

**Fig 1. Location of installation points for SkeeterVac SV3100 traps.** Monitoring area for *Mansonia* spp. mosquitoes, along the side roads adjacent to the Madeira river, Porto Velho, Rondônia, Brazil.

occurrences of low temperatures, around 18˚C. Average rainfall varies between 264mm and 17mm. The rainy season extends from October to April, while the dry season runs from June to August, with transition periods in May and September [17].

The ten SkeeterVac traps were installed in ten homes (one trap per house) in the covered outdoor areas of each of the selected homes, in order to cover the rural, semi-rural and peri-urban environments of the Porto Velho. Mosquitoes were collected from the traps weekly. The study began in June 2019 and ended in January 2023. The mosquito samples were categorized following the epidemiological calendar (1 – 52nd epidemiological week), which enabled a more thorough analysis of the seasonality pattern.

The positioning of the SkeeterVac traps was determined based on three criteria: the presence of human communities, the prevalence of *Mansonia* mosquito attacks, and the presence of potential breeding sites. Satellite images were used to identify breeding sites, which was further confirmed through on-site inspections. To prevent sample overlap, a minimum distance of 5km was maintained between traps.

## SkeeterVac SV3100 automatic traps

This trap consists of an automatic device capable of simulating scent, heat, and light cues for mosquitoes. When attracted, the mosquitoes are vacuumed and stored in a collection drawer, where they are killed by drying out. This device allows the use of pheromones such as Lurex

**Table 1. Georeferences of *Mansonia* spp. monitoring points in Porto Velho.** Showing land use categories and distances (km) from the trap to the nearest breeding site and distance from the trap to the Madeira river.

| Trap code | Locality | Latitude | Longitude | Use of the soil | Distance to the nearest breeding site (km) | Distance to the Madeira River (km) | Sampling duration |
|---|---|---|---|---|---|---|---|
| P01 | Cujubim Grande | -8.585.146 | -63.730.917 | Urban | 1.74 | 0.18 | Jul/19 to Jan/23 |
| P02 | São Domingos | -8.816.567 | -63.990.400 | Agriculture | 0.47 | 0.81 | Jun/19 to Jan/23 |
| P03 | Teotônio | -8.872.650 | -64.053.383 | Urban / Pisciculture | 0.32–1.41 | 0.2 | Jun/19 to Jan/23 |
| P04 | Morrinhos | -9.035.583 | -64.143.650 | Livestock / Agriculture | 2.42 | 3.64 | Jul/19 to Jan/23 |
| P05 | Santa Rita | -9.061.250 | -64.185.450 | Agriculture | 1.2 | 2.95 | Jul/19 to Apr/23 |
| P06 | Jaci Paraná | -9.263.350 | -64.405.033 | Urban | 0.49–0.86 | 5.78 | Jun/19 to Apr/23 |
| P07 | Rio Contra | -9.304.517 | -64.449.933 | Livestock / Agriculture | 0.91 | 11.09 | Jul/19 to Apr/23 |
| P08 | Samauma | -9.246.317 | -64.443.633 | Livestock / Agriculture / Pisciculture | 1.73 | 5.14 | Jul/19 to Jan/23 |
| P09 | Joana D'arc Line 09 | -8.977.217 | -64.318.667 | Livestock | 2.75–4.59 | 6.53 | Sep/19 to Mar/23 |
| P10 | Joana D'arc Line 15 | -9.062.633 | -64.418.083 | Livestock | 3.04 | 9.9 | Sep/19 to Mar/23 |

and Octenol that maximize attractiveness to various species of hematophagous mosquitoes. The trap uses a butane gas cylinder (13 kg), which keeps it operating for approximately 30 uninterrupted days. This way, it is possible to change the cylinder and attractants and keep them collecting mosquitoes for long periods.

In this monitoring, a total of ten locations were studied. Firstly, the locations Jaci Paraná, Teotônio, and São Domingos were the first to have traps installed (June 2019); a month later (July 2019), the locations Cujubim Grande, Morrinhos, Rio Contra, Samauma, and Santa Rita received traps. Then, the locations Joana D'Arc line 09 and Joana D'Arc line 15 were covered with SkeeterVac SV3100 traps (September 2019), however, these traps were retrieved earlier (until March 2020) due to residents' reluctance to participate further in the study, citing legal concerns. In three locations, Jaci Paraná, Santa, and Rio Contra, monitoring was carried out until the end of April 2022, while in other locations, Cujubim Grande, São Domingos, Teotônio, Morrinhos, and Samauma, monitoring was carried out until January 2023 (Table 1).

## Processing and taxonomic identification of mosquitoes

Mosquitoes were always collected in traps on Mondays and Tuesdays at the beginning of each epidemiological week. The specimens were stored in jars with lids containing silica gel, cotton, and filter paper. To prevent the proliferation of fungi, the lids of the jars were sealed with thread-sealing and insulating tape. The pots were stored in a dry environment until taxonomic identification.

The *Mansonia* specimens were identified only to the genus level, as the preservation quality of the mosquitoes upon removal from the trap compartment was insufficient for species-level identification. Male specimens, eventually collected, had their genitalia dissected for specific identification. For taxonomic identification, the dichotomous keys of Consoli and Lourenço-de-Oliveira [1] and Forattini [2] were used. The assembly of male genitalia was based on the protocol described by Sá et al. [18]. The identification of male genitalia was based on the dichotomous keys of Lane [19], Ronderos & Bachmann [20], Cova-Garcia et al. [21], and Belkin et al. [22]. Specimens of *Mansonia*, identified through male genitalia, were deposited in the IEPA entomological collection as voucher material, with the following registration

numbers: #12122 (*Ma. titillans*), #12123 (*Ma. pseudotitillans*), #12124 (*Ma. indubitans*), #12125 (*Ma. amazonensis*) and #12126 (*Ma. humeralis*).

## Data analysis

After identifying the mosquitoes, the data was tabulated in electronic spreadsheets to continue with statistical analyses. Initially, the database was statistically explored to obtain quantity, mean, and frequency metrics per species of mosquitoes and year of sampling. Then, the data was filtered to analyze the same statistical metrics only for the *Mansonia* genus and to verify the seasonality patterns of the genus between locations. Spearman correlations were calculated to check whether the areas showed abundance autocorrelation. The same test was carried out for monthly meteorological variables (temperature, relative humidity, accumulated precipitation, wind speed at 2m height, wind speed at 10m height) and the abundance of *Mansonia*. Finally, we inferred forecasts (104 weeks ahead) of *Mansonia* in the area using time series analysis based on the ARIMA model (AutoRegressive Integrated Moving Average). This analysis was conducted individually for five traps (P01, P02, P03, P04, and P08) (S1 Table), which remained in the field for longer periods. All analyses were performed in R Studio software version 3.0.386 (R Studio Team 2023), using the R language, version 4.2.3 (https://www.R-project.org/), and native packages, such as 'forecast ', 'dplyr', 'ggplot2', 'readxl', and 'zoo'.

## Results

### Richness and abundance of mosquitoes

Over five years of collection, 153,125 mosquitoes were sampled in 10 SkeeterVac SV3100 traps. The genus *Mansonia* was the most abundant, with 82,819 (54.09%) specimens, followed by *Culex* Linnaeus, 1758, with 63,521 (41.48%) and *Coquillettidia* Dyar, 1905 with 3,305 (2.16%). As for location, *Culex* predominated in four points: Cujubim Grande, São Domingos, Santa Rita, and Jaci Paraná, while *Mansonia* was more abundant in six locations: Samauma, Teotônio, Morrinhos, Rio Contra, Joana D'Arc line 09 and Joana D'Arc line 15 (Table 2).

Samauma had the highest relative abundance, with 56,380 specimens collected, while species richness was higher in Rio Contra, with 19 species identified (Table 2). Considering only the genus *Mansonia*, six species were identified by assembling the male genitalia: *Ma. titillans* (Walker, 1848), *Ma. humeralis* Dyar & Knab, 1925, *Ma. amazonensis* (Theobald, 1901), *Ma. flaveola* (Coquillett, 1906) and *Ma. pseudotitillans* (Theobald, 1901). Most of the specimens were identified up to the genus level (*Mansonia* sp.) due to the drying out of the mosquitoes inside the traps, which resulted in damaged specimens. However, the presence of some male specimens made it possible to identify the six species mentioned in Table 2.

Through male genitalia identification, the following locations: São Domingos, Teotônio, Joana D'arc line 09, Cujubim Grande, and Rio Contra, revealed the presence of two *Mansonia* species each, and Morrinho, Santa Rita, Joana D'arc line 15, Samauma, and Jaci-Paraná, exhibited only one species of *Mansonia* each. *Mansonia flaveola* was exclusively found in Joana D'arc line 09, while *Ma. pseudotitillans* was exclusively found in Rio Contra (Table 2).

Throughout the monitored years, a notable increase in the quantity of *Mansonia* was observed in 2020 (n = 23,224–28.0%), with successive declines in the number of mosquitoes captured in 2021 (25.0%), 2022 (19.8%) and in January 2023 (n = 583–13.4%) with a significantly lower amount compared to January 2020 (n = 3241–74.4%). This data becomes evident when comparing the respective years across the ten locations studied, where a greater abundance is observed in the initial two years, followed by a reduction from July 2021 onwards (Fig 2).

**Table 2. Number of mosquitoes collected by species in each location.** The asterisk after the species name indicates that identification was based on male genitalia.

| Species | Cujubim Grande | Jaci Paraná | Morrinhos | Rio Contra | Samauma | Santa Rita | São Domingos | Teotônio | Joana D'arc Line 09 | Joana D'arc Line 15 | Total | % |
|---|---|---|---|---|---|---|---|---|---|---|---|---|
| *Mansonia* sp. | 614 | 1,626 | 13,238 | 4,252 | 35,302 | 2,243 | 21,779 | 2,650 | 225 | 486 | **82,415** | *53.82* |
| *Ma. humeralis** | 1 | 0 | 14 | 0 | 0 | 27 | 250 | 14 | 0 | 0 | **306** | *0.20* |
| *Ma. amazonensis** | 0 | 0 | 0 | 0 | 0 | 0 | 21 | 16 | 0 | 0 | **37** | *0.,02* |
| *Ma. titillans** | 2 | 18 | 0 | 0 | 3 | 0 | 0 | 0 | 4 | 0 | **27** | *0.02* |
| *Ma. indubitans** | 0 | 0 | 0 | 6 | 0 | 0 | 0 | 0 | 0 | 19 | **25** | *0.02* |
| *Ma. flaveola* | 0 | 0 | 0 | 0 | 0 | 0 | 0 | 0 | 8 | 0 | **8** | *0.005* |
| *Ma. pseudotitillans** | 0 | 0 | 0 | 1 | 0 | 0 | 0 | 0 | 0 | 0 | **1** | *0.001* |
| *Culex* sp. | 4,785 | 5,458 | 2,001 | 698 | 20,320 | 4,403 | 23,790 | 1,663 | 123 | 278 | **63,519** | *41.48* |
| *Cx. coronator** | 0 | 0 | 0 | 0 | 0 | 0 | 0 | 0 | 1 | 1 | **2** | *0.001* |
| *Coquillettidia* sp. | 7 | 9 | 35 | 30 | 236 | 23 | 1,514 | 1,405 | 0 | 0 | **3,259** | *2.13* |
| *Cq. venezuelensis* | 1 | 3 | 0 | 12 | 0 | 2 | 22 | 0 | 0 | 0 | **40** | *0.03* |
| *Cq. shannoni* | 0 | 0 | 0 | 2 | 4 | 0 | 0 | 0 | 0 | 0 | **6** | *0.004* |
| *Aedes* sp. | 23 | 545 | 136 | 2 | 37 | 20 | 31 | 16 | 0 | 0 | **810** | *0.53* |
| *Ae. aegypti* | 10 | 682 | 3 | 0 | 6 | 0 | 2 | 3 | 0 | 0 | **706** | *0.,46* |
| *Ae. albopictus* | 1 | 3 | 9 | 0 | 0 | 0 | 1 | 6 | 0 | 0 | **20** | *0.01* |
| *Ae. scapularis* | 0 | 86 | 10 | 2 | 27 | 8 | 0 | 0 | 0 | 0 | **133** | *0.09* |
| *Ae. serratus* | 0 | 0 | 0 | 1 | 0 | 0 | 0 | 0 | 0 | 0 | **1** | *0.001* |
| *Aedeomyia squamipennis* | 2 | 49 | 8 | 27 | 275 | 21 | 344 | 56 | 0 | 0 | **782** | *0.51* |
| *Anopheles* sp. | 17 | 21 | 11 | 140 | 23 | 17 | 137 | 92 | 5 | 0 | **463** | *0.30* |
| *An. darlingi* | 1 | 13 | 0 | 149 | 2 | 3 | 3 | 0 | 1 | 0 | **172** | *0.11* |
| *An. nuneztovari* | 1 | 0 | 0 | 1 | 0 | 0 | 0 | 3 | 0 | 0 | **5** | *0.003* |
| *An. deaneorum* | 0 | 0 | 0 | 0 | 0 | 0 | 0 | 2 | 0 | 0 | **2** | *0.001* |
| *An. triannulatus* | 0 | 0 | 0 | 0 | 1 | 0 | 0 | 0 | 0 | 0 | **1** | *0.001* |
| *Limatus durhamii* | 13 | 13 | 21 | 20 | 83 | 3 | 11 | 10 | 2 | 6 | **182** | *0.12* |
| *Psorophora* sp. | 1 | 14 | 9 | 3 | 10 | 0 | 5 | 3 | 0 | 1 | **46** | *0.03* |
| *Ps. confinnis* | 0 | 2 | 0 | 3 | 0 | 0 | 0 | 0 | 0 | 0 | **5** | *0.003* |
| *Ps. ferox* | 0 | 12 | 0 | 0 | 0 | 0 | 0 | 0 | 0 | 0 | **12** | *0.008* |
| *Uranotaenia* sp. | 12 | 5 | 6 | 2 | 36 | 7 | 21 | 16 | 0 | 0 | **105** | *0.07* |
| *Ur. geometrica* | 0 | 0 | 0 | 0 | 3 | 0 | 0 | 0 | 0 | 0 | **3** | *0.002* |
| *Ur. lowii* | 0 | 0 | 0 | 5 | 1 | 0 | 0 | 0 | 0 | 0 | **6** | *0.004* |
| *Wyeomyia* sp. | 3 | 0 | 0 | 0 | 11 | 0 | 0 | 6 | 2 | 4 | **26** | *0.02* |
| **Total** | **5,494** | **8,559** | **15,501** | **5,356** | **56,380** | **6,777** | **47,931** | **5,961** | **371** | **795** | **153,125** | |
| **%** | *3.6* | *5.6* | *10.1* | *3.5* | *36.8* | *4.4* | *31.3* | *3.9* | *0.2* | *0.5* | | *100* |

The correlogram depicted in Fig 3 shows that at least three groups showed autocorrelation in the abundance of *Mansonia* between different locations. The first group was formed by the two locations of the Joana D'Arc settlement (line 09 and line 15) with the highest Spearman coefficient (s = 0.74). Then, the second group formed by Jaci-Paraná and Santa Rita (s = 0.58), these two locations with Samauma (s = 0.30–0.38), and the three locations with Rio Contra (s = 0.12–0.02). The third group was formed by Morrinho and Teotônio (s = 0.56), these two locations with Cujubim Grande (s = 0.43–0.28). Finally, São Domingos, which did not show a positive relationship with other locations, was recovered as more related to the third group in the dendrogram (Fig 3).

Considering the geographical locations and estimated autocorrelations, the highest correlation coefficient (s = 0.74) observed in the Joana D'Arc settlement can be attributed to the

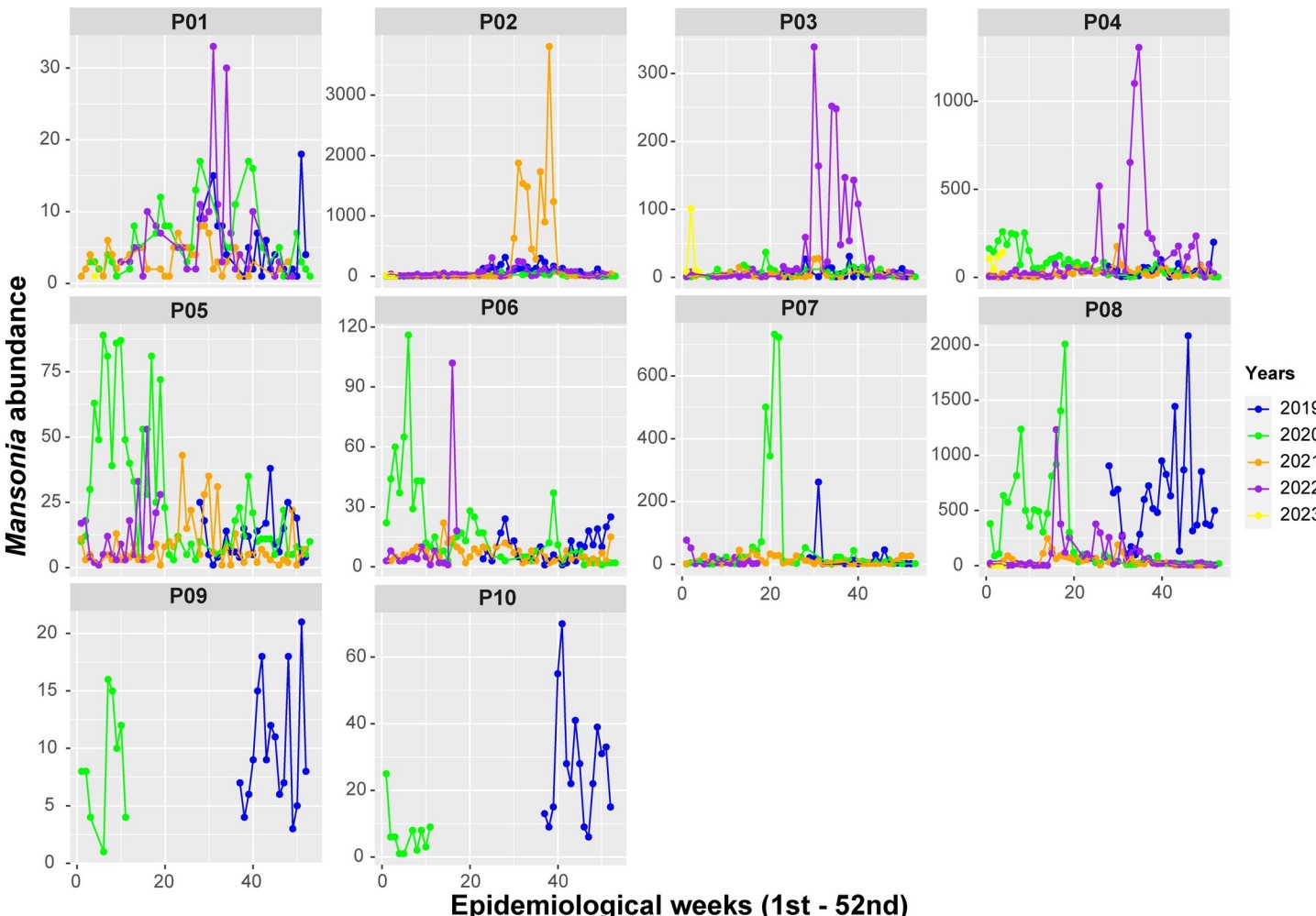

**Fig 2. Annual historical series of *Mansonia* capture with SkeeterVac SV3100 traps, in ten locations in Porto Velho, Rondônia.** Each color represents a year. The y-axis was not standardized to better observe the differences in each location.

proximity between these two collection points (14 km). Similarly, their geographic proximity can justify the correlation between Morrinhos and Teotônio (20 km). Conversely, Cujubim Grande and Teotônio, despite being farther apart (47 km), still exhibited positive autocorrelation. An intriguing finding was observed with São Domingos, which is closer to Teotônio (9.3 km) yet showed null autocorrelation (s = 0), although these two locations are situated on opposite banks of the Madeira River (Fig 3).

During the analysis of environmental factors such as cumulative precipitation, mean relative humidity, average temperature, and wind speed at 2 and 10 meters above ground, we observed that wind speed at 2 meters displayed the highest correlation coefficient (s = 0.68) with the abundance of *Mansonia* in the Joana D'Arc settlement, line 9. São Domingos and Teotônio exhibited a weak positive correlation with average temperature. Teotônio also demonstrated a weak positive correlation with precipitation (Fig 4).

## Time series and forecast

The breakdown and analysis of the historical data series gathered through the SkeeterVac SV3100 over the five years is presented in Fig 5. The peaks of *Mansonia* mosquito abundance

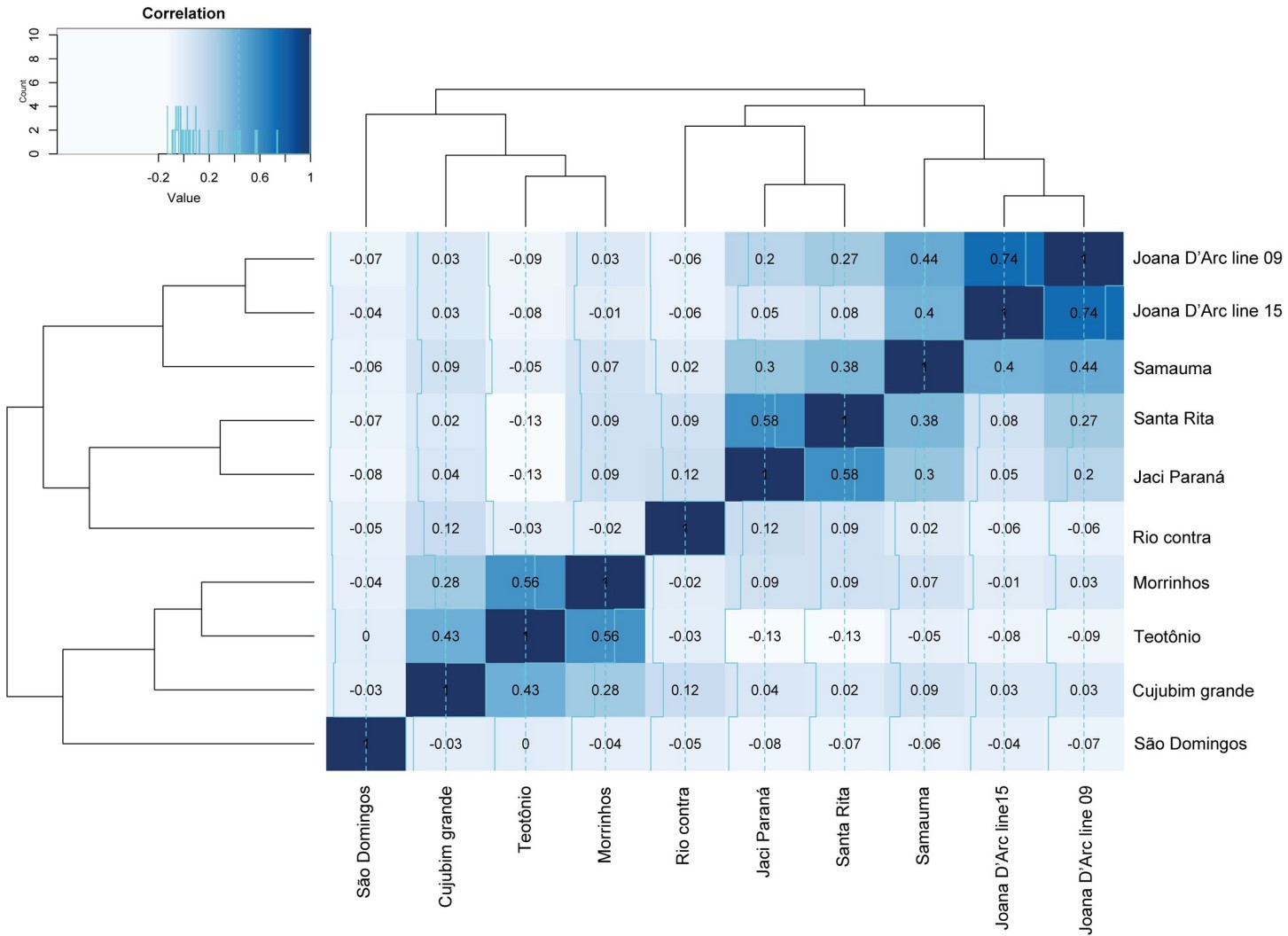

**Fig 3. Spearman correlogram (s) between the abundance of *Mansonia* spp. and the location, with dendrograms estimating the relationship between mosquito collection points.**

were more pronounced in the initial years of the study (2019–2020). Subsequently, isolated peaks in abundance were observed at the beginning of 2022. Thus, the trend indicated a sharp drop at the beginning of 2021, then an increase at the end of 2021, and finally, stabilization with a subsequent drop in abundance in 2023 (Fig 5).

The seasonality estimate for *Mansonia* in Porto Velho was highest in abundance between July and August, with a second, smaller peak in April. The residuals from the analysis showed low variations, between 0 and 2.5 units of collected mosquitoes, indicating that the ARIMA model fit well with the collected data (Fig 5).

When predicting *Mansonia* abundance for the five locations, the model inferred low stability in Cujubim Grande, São Domingos, and Morrinhos. On the other hand, in the Teotônio and Samauma locations, the model inferred stability with a slight upward trend in the number of *Mansonia* for the next two years (2024 and 2025). However, we highlight that the RMSE values of 201.5, MAE of 77.4, and MAPE of 9.2 reveal a lot of inaccuracy for this model (Fig 6).

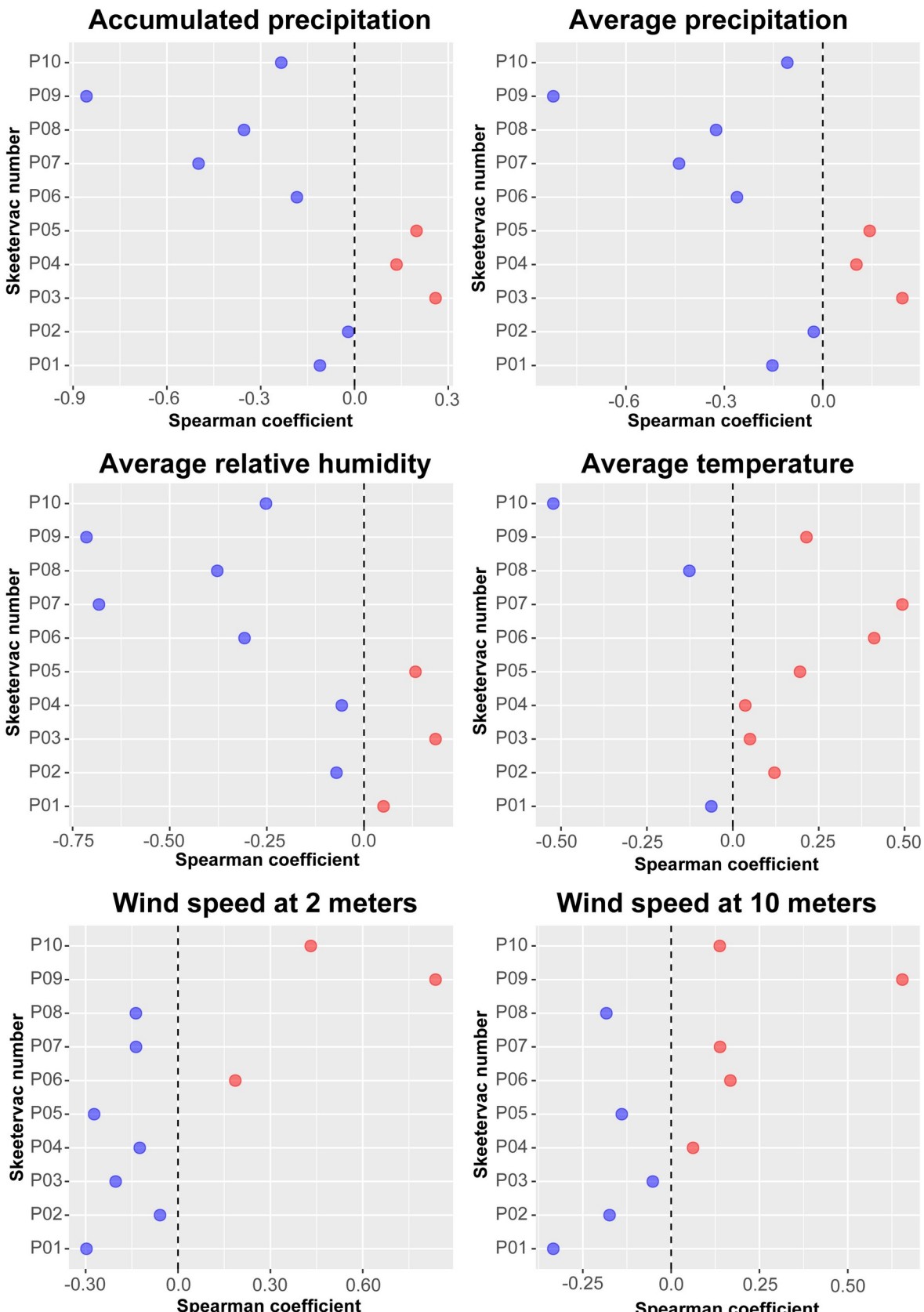

**Fig 4. Correlation of the abundance of *Mansonia* spp. in the teen locations with six environmental variables obtained from the SAE meteorological station.** Blue dots represent negative correlation and red dots represent positive correlation with abundance.

## Discussion

In this study, 24 species of Culicidae were identified in ten entomological monitoring points, all previously identified in previous studies in Porto Velho [11, 12, 23–25]. The genus *Mansonia* was the most abundant in the study and is known to cause nuisance to surrounding human populations. Although this group of mosquitoes is not yet considered a competent disease vector in Brazil [11], field tests revealed the positivity of some specimens for the Mayaro virus [26], indicating the need for further investigation into its epidemiological importance for the region.

In four locations, the genera *Culex* and *Aedes* were the most frequently collected, showing a statistically significant predominance in some of them. Notably, these groups of mosquitoes were not correlated with dams or breeding sites linked to the hydroelectric plant; on the contrary, they demonstrated a strong association with anthropic environments [1, 2, 27]. These

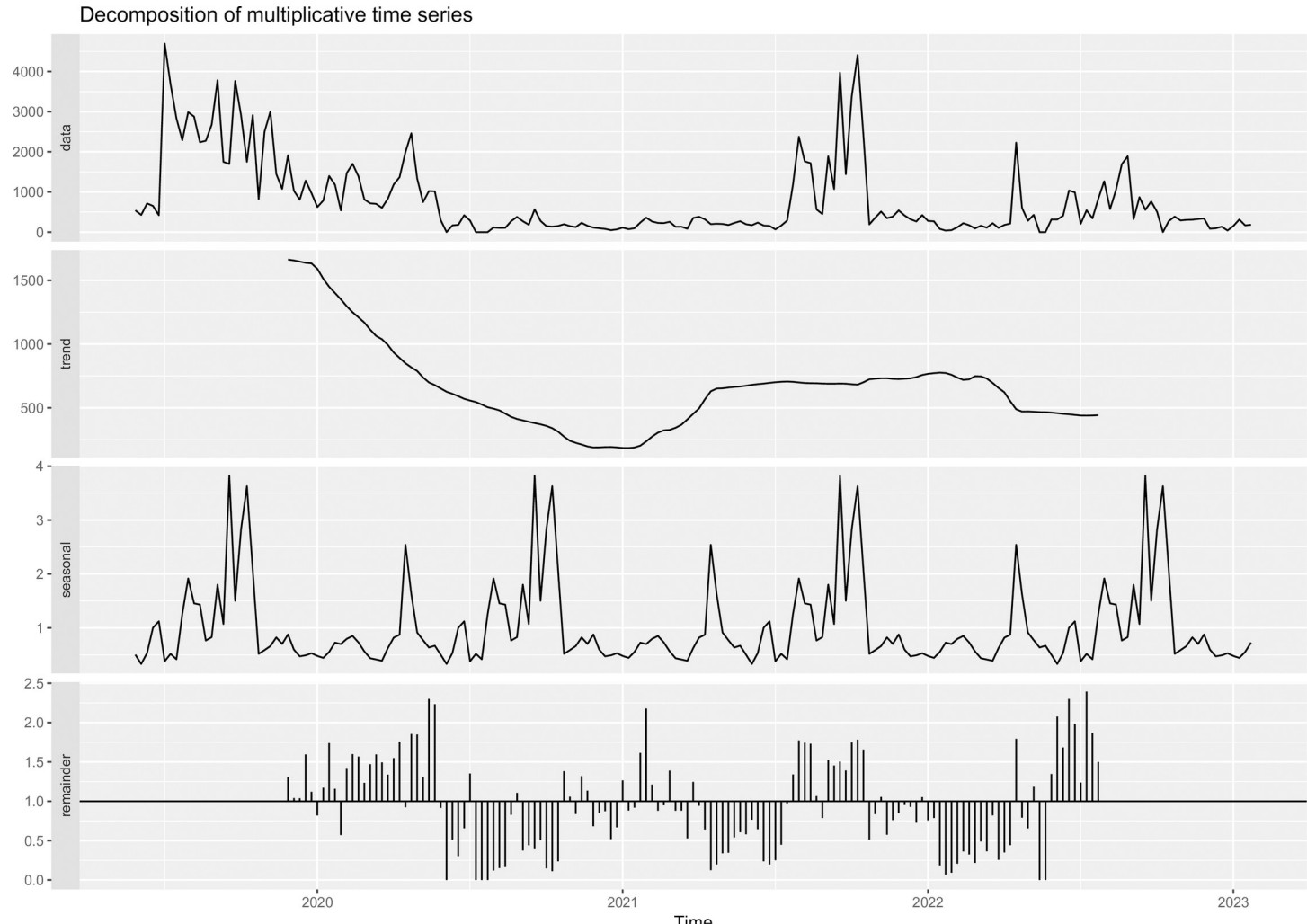

**Fig 5. Time series decomposition to estimate the seasonality of *Mansonia* mosquitoes in five in the region of Porto Velho, Rondônia.**

## Forecast for P01

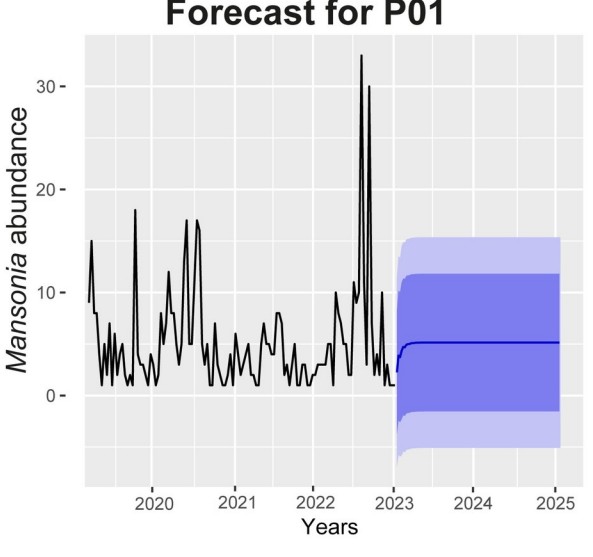

## Forecast for P02

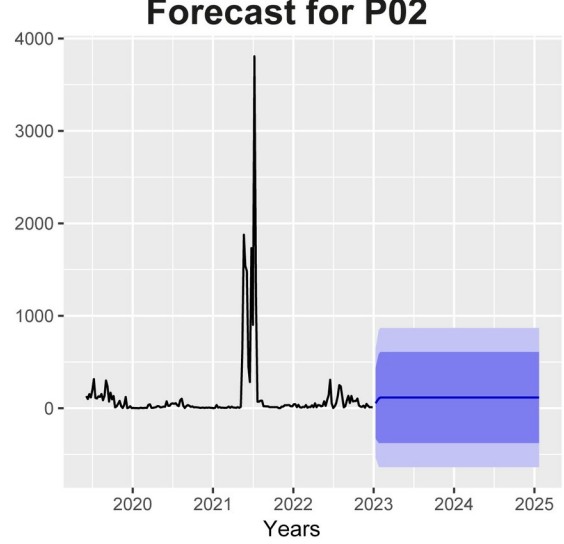

## Forecast for P03

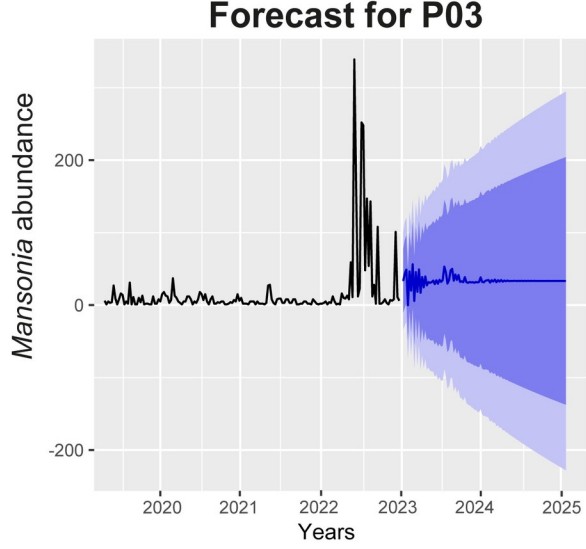

## Forecast for P04

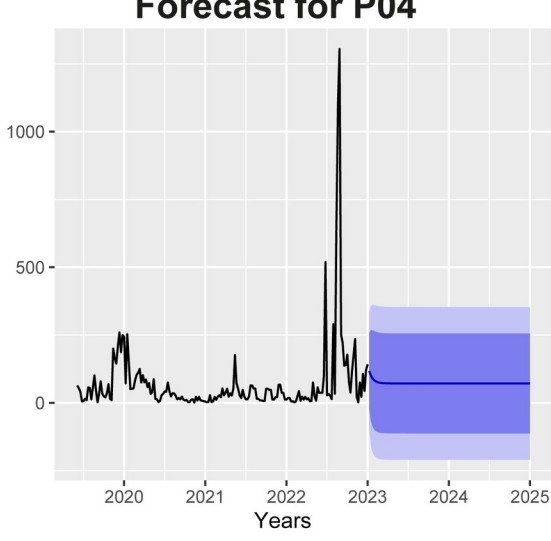

## Forecast for P08

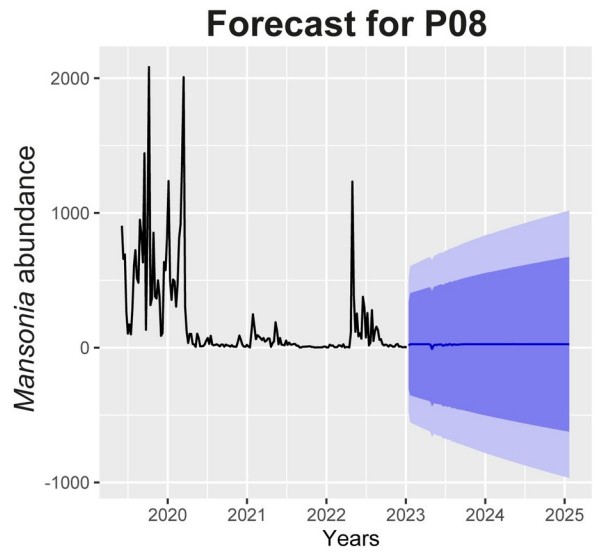

**Fig 6. Forecast of future abundance of *Mansonia* for the next three years based on the ARIMA model of past data collected with SkeeterVac SV3100 traps in Porto Velho, Rondônia.** P01 –Cujubim Grande, P02 –São Domingos, P03 –Teotônio, P08 –Samauma.

mosquitoes find favorable conditions for reproduction in mobile deposits or water storage tanks, often maintained by the community itself. Furthermore, *Aedes aegypti* (Linnaeus, 1762) and *Aedes albopictus* (Skuse, 1895) are invasive species in the Amazon region [28], and they are closely involved in the transmission of diseases, such as arboviruses and microfilariae [1].

On the other hand, the genus *Mansonia* is a group of native mosquitoes, widely distributed in the Amazon region, revealing a remarkable ability to adapt to the complex climatic conditions that have persisted over millions of years [29]. In this region, several species of mosquitoes of significant relevance coexist, such as *Anopheles darlingi* Root, 1926, responsible for the transmission of malaria [25], and *Haemagogus janthinomys* Dyar, 1921, vector of wild yellow fever [30]. Additionally, the presence of invasive species *Aedes aegypti* and *Aedes albopictus* amplifies the complexity of this panorama [31]. Given this scenario, studies that investigate the factors that drive the increase in the abundance of these mosquitoes and the invasion of exotic species are necessary. Among the elements documented with significant influence, deforestation stands out, with changes in land use [32], climate change [33, 34], and the introduction of livestock or domestic animals, which serve as sources of blood meal [35]. In this context, entomological monitoring is indispensable for conducting in-depth investigations into these phenomena.

Entomological monitoring with SkeeterVac SV3100 automatic traps proved effective in capturing *Mansonia*, being attractive through pheromones (Lurex and Octenol) and light, heat, and $CO_2$ signals. Originally developed to reduce the rate of blood-sucking insect bites near homes [36], the SkeeterVac SV3100 stood out for its ease of handling and the ability to carry out long uninterrupted collections. However, the main negative points were challenges such as monthly gas recharge and baits with high import value for Brazil. Additionally, the prolonged stay (one week) of the mosquitoes in the trap resulted in the specimens drying out, causing damage to the morphological identification structures. This last limitation is crucial in contexts where morphological preservation is necessary. However, in studies that do not require more specific identification or in cases where molecular techniques can be used for identification, this trap can be a practical tool for monitoring mosquitoes.

Furthermore, it is crucial to consider an additional aspect related to the efficacy of Skeeter-Vac traps: the protection they provide is not solely dependent on the attraction rate and capture efficiency but also on the reduction of bites experienced by the residents in trap-utilizing homes. Despite numerous studies reporting substantial mosquito captures, extended usage of these traps for control purposes has not exhibited a significant impact [37, 38]. This is partially attributed to the necessity for a sustained reduction in the influx of new mosquitoes over time. This phenomenon is particularly noticeable in the case of *Mansonia*, where the presence of numerous breeding sites close to monitored areas sustains high mosquito populations. This observation is supported by the fluctuations in mosquito numbers observed throughout the study. Despite these variations, complete elimination within residences was not attained, emphasizing the utility of these traps as an effective instrument for entomological monitoring and not as a method for mosquito control.

Our entomological monitoring revealed that the areas most prone to abundance of *Mansonia* spp. are located close to cattle farms, areas of forest deforestation, small lakes, and shallow backwaters, densely covered with macrophytes, which provide ideal conditions for the creation of immature forms and blood food for adult females [12]. Additionally, the analysis of the historical series of collected data indicated a tendency for a reduction over the years analyzed,

especially after 2021. This decrease is also associated with activities to remove macrophytes from the banks of the Madeira River, backwaters, and shallow lakes near the area that comprises the Santo Antônio hydroelectric plant dam, considering that the removal of such plants through mechanical, biological, or chemical control effectively prevents the development of immature forms of these mosquitoes [39].

Regarding correlations in *Mansonia* abundance across the investigated areas, we observed that local environmental characteristics may be responsible for the correlation among the monitoring points. For example, in rural areas, the primary contributors to the blood sources were predominantly cattle, chickens, and pigs, while in urban areas, dogs, humans, and cats predominated. This diversity in the blood source composition implies the potential existence of different species, each demonstrating specific blood-feeding preferences [35]. Nevertheless, it is imperative to emphasize that further investigations are required to substantiate our hypothesis.

Spearman's analysis between *Mansonia* abundance and environmental variables resulted in very discrepant indices between locations. This discrepancy may be associated with some monitoring points being very far from the meteorological station. Wind speed emerged as the most important variable when considering this. Intuitively, wind may play a significant role in the passive transport of mosquitoes between distant locations [40, 41]. In studies carried out in Malaysia using the mark-release-recapture method, the recapture of *Mansonia* varied between 0.5 and 2.4 km from its release points [42].

Furthermore, a field experiment conducted by de Mello et al. [43] in Porto Velho concluded that the dispersal movement of *Mansonia* is predominantly carried out by short and random flights, generally below 1 m in height, maintaining a home range within a radius of approximately 30 to 100 m from the point of emergence of the adults. This highlights a tendency for these mosquitoes to remain close to breeding sites in specific vegetation fragments. Although the maximum recapture distance for mosquitoes was 2km, their general tendency is to stay on the same patch of vegetation under stable conditions of food sources and environmental preservation [43].

The seasonality inferred by the ARIMA model for *Mansonia* breakdown into two abundance peaks, one in July and a smaller one in August of each year. Inference on future abundance was recovered with several limitations and high RMSE, MAE, and MAPE indices, revealing that the inference obtained does not present statistical significance but only approximates a likely future scenario. Extreme events can also impact the forecasts obtained. For example, the historic flood of 2014 likely increased the abundance of *Mansonia* by flooding areas inaccessible by the typical flood and flow patterns of the Madeira River [44, 45]. In this scenario of historic floods, aquatic macrophytes can be carried to these new environments and form new breeding sites for *Mansonia*, generating locations that are increasingly remote and inaccessible for control.

Faced with historic droughts, such as the one we experienced in 2023 due to the El Niño phenomenon [46], environments with macrophytes are prone to drying out, eliminating plants and, consequently, immature forms of *Mansonia* spp. This scenario suggests a potential for reducing the local abundance of mosquitoes. However, in light of these observations, it is crucial to emphasize the importance of more detailed investigations on the seasonal behavior of *Mansonia* species. This study highlights the pressing need for such research to be continued to achieve a deeper and more comprehensive understanding of the ecological dynamics of these organisms in response to climate change. A dedicated commitment to this ongoing approach is essential for updating environmental control and management strategies, ensuring their enduring efficacy and sustainability. This perspective holds significance from both academic and regulatory perspectives.

## Supporting information

**S1 Table. Number of mosquitoes collected weekly with SkeeterVac traps.** The forecast analyses were carried out only with traps that remained in the field for a longer period: P01, P02, P03, P04 and P08.
(PDF)

## Acknowledgments

We thank Dr. Maria Anice Mureb Sallum for reviewing the identifications of the slides with the mosquito's genitalia, Noel Neto F. dos Santos, Rafaela Andressa, and Raimundo Nonato Mendes for inspecting the traps, and Dayse Swelen Ferreira, for creating the map. We want to thank the financial support through a post-doctoral scholarship from the Fundação para o Desenvolvimento da Universidade Estadual Paulista–FUNDUNESP granted to José F. Saraiva. We also thank @Stylus–California–USA (https://www.linkedin.com/company/atstylus/) for English language editing.

## Author Contributions

**Conceptualization:** José Ferreira Saraiva, Allan Kardec Ribeiro Galardo, José Bento Pereira Lima.

**Data curation:** José Ferreira Saraiva.

**Formal analysis:** José Ferreira Saraiva, Ahana Maitra.

**Funding acquisition:** Dario P. Carvalho, Allan Kardec Ribeiro Galardo, José Bento Pereira Lima.

**Investigation:** José Ferreira Saraiva, Nercy Virginia Rabelo Furtado.

**Project administration:** Allan Kardec Ribeiro Galardo, José Bento Pereira Lima.

**Resources:** José Bento Pereira Lima.

**Supervision:** José Bento Pereira Lima.

**Validation:** José Bento Pereira Lima.

**Writing – original draft:** José Ferreira Saraiva.

**Writing – review & editing:** José Ferreira Saraiva, Nercy Virginia Rabelo Furtado, Ahana Maitra, Dario P. Carvalho, Allan Kardec Ribeiro Galardo, José Bento Pereira Lima.

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
