## [Decision Letter · Decision Letter 0]

1 Feb 2024

PONE-D-23-44055Seasonality of Mansonia Blanchard (Diptera, Culicidae, Mansoniini)in Porto Velho, State of Rondônia, Brazil.PLOS ONE

Dear Dr. Saraiva,

Thank you for submitting your manuscript to PLOS ONE. After careful consideration, we feel that it has merit but does not fully meet PLOS ONE’s publication criteria as it currently stands. Therefore, we invite you to submit a revised version of the manuscript that addresses the points raised during the review process.

Both the reviewers have made a substantial amount of comments and suggestions, please address them individually. Also, please note that there might be some potential conflict of interest to clarify.

We look forward to receiving your revised manuscript.

Kind regards,

Luca Nelli, PhD

Academic Editor

PLOS ONE

Journal Requirements:

"Research and Development project from Santo Antônio Energia (ANEEL project CT.PD.124.2018)."

5. We note that your Data Availability Statement is currently as follows: All relevant data are within the manuscript and its Supporting Information files.

7. We note that Figure 1 in your submission contain map/satellite images which may be copyrighted. All PLOS content is published under the Creative Commons Attribution License (CC BY 4.0), which means that the manuscript, images, and Supporting Information files will be freely available online, and any third party is permitted to access, download, copy, distribute, and use these materials in any way, even commercially, with proper attribution. For these reasons, we cannot publish previously copyrighted maps or satellite images created using proprietary data, such as Google software (Google Maps, Street View, and Earth). For more information, see our copyright guidelines: http://journals.plos.org/plosone/s/licenses-and-copyright.

Reviewers' comments:

Reviewer's Responses to Questions

**Comments to the Author**

1. Is the manuscript technically sound, and do the data support the conclusions?

Reviewer #1: Yes

Reviewer #2: Yes

2. Has the statistical analysis been performed appropriately and rigorously? 

Reviewer #1: Yes

Reviewer #2: Yes

3. Have the authors made all data underlying the findings in their manuscript fully available?

Reviewer #1: Yes

Reviewer #2: No

4. Is the manuscript presented in an intelligible fashion and written in standard English?

Reviewer #1: Yes

Reviewer #2: No

5. Review Comments to the Author

Reviewer #1: Article revisions PLOS ONE

Title: Seasonality of Mansonia Blanchard (Diptera, Culicidae, Mansoniini) in Porto Velho, State of Rondônia, Brazil, is relevant and is in line with the scope of the Journal. However, the following modifications need to be addressed to make it publishable:

Manuscript Number: PONE-D-23-44055

Abstract

Pg. 2. First sentence is too long and could be better written to improve comprehension. Suggestion: “Entomological research is vital for shaping strategies to control mosquito vectors. Its significance also reaches into environmental management, aiming to prevent inconveniences caused by non-vector mosquitoes like the Mansonia Blanchard mosquito.”

Pg. 2. Where it says: “In this study, we carried out a five-year comprehensive monitoring..” I would recommend writing the years of study in parenthesis. E.g.: “…we carried out a five-year (2019-2023)…”.

Pg. 2. Second sentence. There is too much repetition of the word location. Suggestion: “In this study, we conducted a comprehensive five-year (2019-2023) monitoring of these mosquitoes at ten sites in Porto Velho, Rondônia, using SkeeterVac SV3100 automatic traps positioned within the Madeira River hydroelectric complex..”

Pg. 2. Fourth sentence. “Notably, a significant decreasing trend in local abundance was

observed throughout the study.” It would be interesting to inform which months or year was this significant decrease in the density of Mansonia spp.

Introduction

Pg. 3. First sentence: “Entomological monitoring of mosquitoes represents an important part of public health data collection, as many of these insects are vectors of several human diseases.” Mosquitoes are vector of pathogens not diseases. Suggestion: “Entomological monitoring of mosquitoes represents an important part of public health data collection, as many of these insects are vectors of several human pathogens.”

Pg. 3. Delete the “ – “ symbol at the middle of the third sentence of the paragraph.

Material and methods

Area of the study

Indicate in the Methods section the license number of the collection.

What was the criterion used to decide on the distances between the collection areas?

The authors emphasize the importance of selecting a study area for comprehensive point sampling.

Add sources for demography.

In which entomological collection were the collected specimens deposited? Please provide this information along with their deposit numbers.

However, there are concerns about the methodology. Mosquitoes were extracted weekly from traps with a high specimen abundance, causing damage and compromised identifications. It's unclear why specimens weren't removed sooner. Identifying species richness is challenging, especially as many specimens were only identified to the genus level (Mansonia sp.) due to drying out in traps.

Figure 1. The legend of the map shows the color of Brazil as white, however the country appears as green, please correct it.

Pg. 5. First line of the third paragraph. Where it says: “The automatic traps were installed at ten points, each positioned in the surroundings of the selected residences.” How many residencies? Earlier on it mentions that the locations chosen for the study are characterized as rural, semi-rural, or peri-urban environments, how many traps were placed in each one of these classifications of level of urbanization?

Pg. 5. It is not mandatory, however, since the SkeeterVac SV3100 Automatic Traps are a new technology not used extensively, it would be more informative to add a picture of this trap and describe better its different parts, giving more details about its design and structure.

Pg. 5. It appears the traps were installed in different time periods in each location. Ideally for a correct statistical analysis all locations, to be compared to one another, should have the same period. How is it possible to compare July 2019 from locations: Jaci Paraná, Teotônio, and São Domingos if you don´t have data from the same month and year from the other locations (e.g. Cujubim Grande, Morrinhos, Rio Contra, Samauma, and Santa Rita)?

Pg. 6 Second paragraph under “Processing and taxonomic identification of Mosquitoes” says: “Taxonomic determination took place down to the genus level due to the low quality of the specimens when they were removed from the trap compartment.” Than latter on in the text under “Data Analysis” it says: “Initially, the database was statistically explored to obtain quantity, mean, and frequency metrics per species of mosquitoes…”. It could not have been “per species” because they were only identified to genus level right? Please correct it.

Pg. 8. Second paragraph. Please revise the presence of em dash "—" in this paragraph, is it necessary or was it a typing error. Also, where it says: “…revealed the presence of two Mansonia species each, and Morrinho, Santa Rita Joana D'arc line 15, Samauma..” is “line 15” part of the name of the location? If not, please delete it.

Table 2 lacks clarity and doesn't quantify identified species, hindering understanding. Clarifications in the table are needed for a more straightforward presentation of the study's findings.

Pg. 9 The paragraph starting with “Considering the geographic location…” Seems to belong in the discussion part of the manuscript, since it justifies the results obtained. Consider if this is a change worth making.

Pg 9. Where it says: “…obtained in the Joana D’Arc settlement is justified by the geographic proximity between these two collection points (14 km). The same can be said between Morrinhos and Teotônio (20 km).” Were these distances at random? I don´t recall seeing the trap distances in the Material and Methods section of the manuscript.

Pg. 9 Under “Time series and forecast” where it says: “The decomposition of the historical series….” A better word choice could be made in this sentence. Suggestion: “The breakdown and analysis of the historical data series gathered through the SkeeterVac SV3100 over a five-year period is presented in Figure 5.”

Pg 9. Second sentence of the last paragraph on page nine. Could be improved. Suggestion: “The peaks of Mansonia mosquitoes' abundance were more pronounced in the initial years of the study (2019-2020).”

Pg 11. Correct “CO2” to “CO2”.

Discussion

The discussion is extensive, however, lacks important results obtained in the study. It does not discuss the results regarding wind speed being the most relevant meteorological factor influencing the abundance of Mansonia.

Pg 11. Last sentence of the last paragraph on page ten that goes on to page eleven. Where it says: “Furthermore, Aedes aegypti (Linnaeus, 1762) and Aedes albopictus (Skuse, 1895) are invasive species in the Amazon region..” Were these species collected in this study?

Pg. 11. First paragraph mentions Anopheles darlingi, Haemagogus janthinomys, Aedes aegypti and Aedes albopictus, were all these species collected in this study? If not, it would be important to focus on Mansonia and not other species. At the end of the parahraph it mentions deforestation, how do the results from the different levels of urbanization mentioned at the beginning (rural, semi-rural, or peri-urban environments) play a role in mosquito abundance? It would be interesting to mention this here.

The authors could insert the paragraph that starts with: “Our entomological monitoring revealed that the areas most prone to abundance of Mansonia spp. are located close to cattle farms, areas of forest deforestation, small lakes…” Right after this one, it seems that this is the information that was lacking.

Reviewer #2: The manuscript needs to be reviewed by either an editing company or by an English native speaker.

In addition, authors must either deposit their data set in Dryad or other platform that gives a doi number for the data set.

Authors declared no conflict of interest. However, their study was financially supported by Research and Development project from Santo Antônio Energia (ANEEL project CT.PD.124.2018). Also, the field collections were carried out in locations in the vicinities of the reservoir lake of the same company that gave them the financial support Santo Antônio (SAE). They should clarify this important conflict.

Statistical analyses are adequate and defined based on the major objectives of the study and data they have at hand. The number of mosquitoes they collected in five years using 10 traps and a few locations is impressive (153,125 mosquitoes were sampled in 10 SkeeterVac

SV3100 traps. The genus Mansonia was the most abundant, with 82,819 (54.09%) specimens, followed by Culex with 63,521 (41.48%) and Coquillettidia with 3,305 (2.16%).

It is not clear why authors did not separate the species in the study and in Table 2. If they identified a certain number of species in each genus; I assume they know which species they have at hand. I suggest they redo the analysis using the species level for Mansonia (the focus of the study) as far as possible, keeping the genus analyses they already did. The stud will be sounder and bring to light new information on species associated with lake reservoirs of hydroelectric power plants.

6. PLOS authors have the option to publish the peer review history of their article (what does this mean?). If published, this will include your full peer review and any attached files.

Reviewer #1: No

Reviewer #2: No

---

## [Author Response · Author response to Decision Letter 0]

17 Mar 2024

Editor

Answer: done.

Answer: done.

Answer: done.

"Research and Development project from Santo Antônio Energia (ANEEL project CT.PD.124.2018)."

Answer: done.

5. We note that your Data Availability Statement is currently as follows: All relevant data are within the manuscript and its Supporting Information files.

Answer: done.

Answer: done.

7. We note that Figure 1 in your submission contain map/satellite images which may be copyrighted. All PLOS content is published under the Creative Commons Attribution License (CC BY 4.0), which means that the manuscript, images, and Supporting Information files will be freely available online, and any third party is permitted to access, download, copy, distribute, and use these materials in any way, even commercially, with proper attribution. For these reasons, we cannot publish previously copyrighted maps or satellite images created using proprietary data, such as Google software (Google Maps, Street View, and Earth). For more information, see our copyright guidelines: http://journals.plos.org/plosone/s/licenses-and-copyright.

 Answer: adjusted.

Answer: checked.

Reviewer #1: 

Abstract

Pg. 2. First sentence is too long and could be better written to improve comprehension. Suggestion: "Entomological research is vital for shaping strategies to control mosquito vectors. Its significance also reaches into environmental management, aiming to prevent inconveniences caused by non-vector mosquitoes like the Mansonia Blanchard mosquito."

Answer: suggestion accepted.

Pg. 2. Where it says: "In this study, we carried out a five-year comprehensive monitoring.." I would recommend writing the years of study in parenthesis. E.g.: "…we carried out a five-year (2019-2023)…".

Answer: done.

Pg. 2. Second sentence. There is too much repetition of the word location. Suggestion: "In this study, we conducted a comprehensive five-year (2019-2023) monitoring of these mosquitoes at ten sites in Porto Velho, Rondônia, using SkeeterVac SV3100 automatic traps positioned within the Madeira River hydroelectric complex."

Answer: done.

Pg. 2. Fourth sentence. "Notably, a significant decreasing trend in local abundance was

observed throughout the study." It would be interesting to inform which months or year was this significant decrease in the density of Mansonia spp.

Answer: information inserted in the sentence.

Introduction

Pg. 3. First sentence: "Entomological monitoring of mosquitoes represents an important part of public health data collection, as many of these insects are vectors of several human diseases." Mosquitoes are vector of pathogens not diseases. Suggestion: "Entomological monitoring of mosquitoes represents an important part of public health data collection, as many of these insects are vectors of several human pathogens."

Answer: change accepted.

Pg. 3. Delete the "– "symbol at the middle of the third sentence of the paragraph.

Answer: done.

Material and methods

Area of the study

Indicate in the Methods section the license number of the collection.

Answer: done.

What was the criterion used to decide on the distances between the collection areas?

The authors emphasize the importance of selecting a study area for comprehensive point sampling.

Answer:

Add sources for demography.

Answer: done.

In which entomological collection were the collected specimens deposited? Please provide this information along with their deposit numbers.

Answer: Entomological collection of the Instituto de Pesquisas Científicas e Tecnológicas do Estado do Amapá - IEPA. Deposit voucher numbers were inserted into the manuscript.

However, there are concerns about the methodology. Mosquitoes were extracted weekly from traps with a high specimen abundance, causing damage and compromised identifications. It's unclear why specimens weren't removed sooner. Identifying species richness is challenging, especially as many specimens were only identified to the genus level (Mansonia sp.) due to drying out in traps.

Answer: The decision to conduct a weekly inspection of the traps was motivated mainly by the significant distance between them, as exemplified by the distance of 155 km between Cujubim Grande and Jaci Paraná. Moreover, most traps were installed on unpaved rural roads, making it even more challenging to move between points to remove mosquitoes. Faced with these adversities and considering the limited resources available to cover the extensive distance, we inspected the traps weekly.

It should be noted that this activity was carried out by two technicians using one vehicle. Each inspection took the entire day, starting at 7:00 in the morning and ending at 6:00 in the evening. Although we recognize that this prolonged time interval may have resulted in damage to the specimens, we emphasize that this was the most feasible and viable period to cover the entire area of interest of the study.

Figure 1. The legend of the map shows the color of Brazil as white, however the country appears as green, please correct it.

Answer: adjusted.

Pg. 5. First line of the third paragraph. Where it says: "The automatic traps were installed at ten points, each positioned in the surroundings of the selected residences." How many residencies? Earlier on it mentions that the locations chosen for the study are characterized as rural, semi-rural, or peri-urban environments, how many traps were placed in each one of these classifications of level of urbanization?

Answer: A total of 10 traps were used. Each trap was installed in the peri-domestic area of the selected residence, i.e., 10 residences. To summarize the number of traps per environment, we provided a table showing the geographic location of the traps, land use, and straight-line distances from water bodies closest to the point.

Pg. 5. It is not mandatory, however, since the SkeeterVac SV3100 Automatic Traps are a new technology not used extensively, it would be more informative to add a picture of this trap and describe better its different parts, giving more details about its design and structure.

Answer: We understand the importance of clarifying the design of the SkeeterVac trap. Therefore, we have improved the section that describes the trap in detail and included specific references to the design used. We believe this information is comprehensive enough so anyone interested in the trap can easily find and purchase it.

Pg. 5. It appears the traps were installed in different time periods in each location. Ideally for a correct statistical analysis all locations, to be compared to one another, should have the same period. How is it possible to compare July 2019 from locations: Jaci Paraná, Teotônio, and São Domingos if you don't have data from the same month and year from the other locations (e.g. Cujubim Grande, Morrinhos, Rio Contra, Samauma, and Santa Rita)?

Answer: Comparisons between locations were conducted, taking into account the same time interval in which the traps were in operation. In the case of traps installed before and that remained in operation until the end of the study, when comparing them with traps installed later, we proceeded with cutting the initial data from the first trap. This same approach was applied to traps that were removed after the start of the study, with the final data excluded from the analyses.

Pg. 6 Second paragraph under "Processing and taxonomic identification of Mosquitoes" says: "Taxonomic determination took place down to the genus level due to the low quality of the specimens when they were removed from the trap compartment." Than latter on in the text under "Data Analysis" it says: "Initially, the database was statistically explored to obtain quantity, mean, and frequency metrics per species of mosquitoes…". It could not have been "per species" because they were only identified to genus level right? Please correct it.

Answer: adjusted.

Pg. 8. Second paragraph. Please revise the presence of em dash "—" in this paragraph, is it necessary or was it a typing error. Also, where it says: "…revealed the presence of two Mansonia species each, and Morrinho, Santa Rita Joana D'arc line 15, Samauma.." is "line 15" part of the name of the location? If not, please delete it.

Answer: adjusted.

Table 2 lacks clarity and doesn't quantify identified species, hindering understanding. Clarifications in the table are needed for a more straightforward presentation of the study's findings.

Answer: The table has been corrected to show all species identified in the study.

Pg. 9 The paragraph starting with "Considering the geographic location…" Seems to belong in the discussion part of the manuscript, since it justifies the results obtained. Consider if this is a change worth making.

Answer: adjusted.

Pg 9. Where it says: "…obtained in the Joana D'Arc settlement is justified by the geographic proximity between these two collection points (14 km). The same can be said between Morrinhos and Teotônio (20 km)." Were these distances at random? I don't recall seeing the trap distances in the Material and Methods section of the manuscript.

Answer: Distance values refer to straight-line measurements. We have included further clarifications about this measure in the text.

Pg. 9 Under "Time series and forecast" where it says: "The decomposition of the historical series…." A better word choice could be made in this sentence. Suggestion: "The breakdown and analysis of the historical data series gathered through the SkeeterVac SV3100 over a five-year period is presented in Figure 5."

Answer: "decomposition" is a term generally used for time series analyses. However, we will accept the change to "breakdown".

Pg 9. Second sentence of the last paragraph on page nine. Could be improved. Suggestion: "The peaks of Mansonia mosquitoes' abundance were more pronounced in the initial years of the study (2019-2020)."

Answer: done.

Pg 11. Correct "CO2" to "CO2".

Answer: done. CO2

Discussion

The discussion is extensive, however, lacks important results obtained in the study. It does not discuss the results regarding wind speed being the most relevant meteorological factor influencing the abundance of Mansonia.

Answer: We consider the discussion to be of good size. Furthermore, we discuss the wind results on line 359, with results found by Alencar et al. 2021, who carried 

---

## [Decision Letter · Decision Letter 1]

1 Apr 2024

PONE-D-23-44055R1Trends of Mansonia (Diptera, Culicidae, Mansoniini) in Porto Velho: seasonal patterns and meteorological influencesPLOS ONE

Dear Dr. Saraiva,

Thank you for submitting your manuscript to PLOS ONE. After careful consideration, we feel that it has merit but does not fully meet PLOS ONE’s publication criteria as it currently stands. Therefore, we invite you to submit a revised version of the manuscript that addresses the points raised during the review process.

 Please make sure to address the remaining comments from a reviewer:

<<The authors did not respond to the following question. What was the criterion used to decide on the distances between the collection areas? The authors emphasize the importance of selecting a study area for comprehensive point sampling.

>>Please submit your revised manuscript by May 16 2024 11:59PM. If you will need more time than this to complete your revisions, please reply to this message or contact the journal office at plosone@plos.org. Please include the following items when submitting your revised manuscript:A rebuttal letter that responds to each point raised by the academic editor and reviewer(s). You should upload this letter as a separate file labeled 'Response to Reviewers'.A marked-up copy of your manuscript that highlights changes made to the original version. You should upload this as a separate file labeled 'Revised Manuscript with Track Changes'.An unmarked version of your revised paper without tracked changes. You should upload this as a separate file labeled 'Manuscript'.If applicable, we recommend that you deposit your laboratory protocols in protocols.io to enhance the reproducibility of your results. Protocols.io assigns your protocol its own identifier (DOI) so that it can be cited independently in the future. For instructions see: https://journals.plos.org/plosone/s/submission-guidelines#loc-laboratory-protocols. Additionally, PLOS ONE offers an option for publishing peer-reviewed Lab Protocol articles, which describe protocols hosted on protocols.io. Read more information on sharing protocols at https://plos.org/protocols?utm_medium=editorial-email&utm_source=authorletters&utm_campaign=protocols.

We look forward to receiving your revised manuscript.

Kind regards,

Luca Nelli, PhD

Academic Editor

PLOS ONE

Journal Requirements:

Reviewers' comments:

Reviewer's Responses to Questions

**Comments to the Author**

1. If the authors have adequately addressed your comments raised in a previous round of review and you feel that this manuscript is now acceptable for publication, you may indicate that here to bypass the “Comments to the Author” section, enter your conflict of interest statement in the “Confidential to Editor” section, and submit your "Accept" recommendation.

Reviewer #1: (No Response)

Reviewer #3: All comments have been addressed

2. Is the manuscript technically sound, and do the data support the conclusions?

Reviewer #1: Yes

Reviewer #3: Yes

3. Has the statistical analysis been performed appropriately and rigorously? 

Reviewer #1: Yes

Reviewer #3: Yes

4. Have the authors made all data underlying the findings in their manuscript fully available?

Reviewer #1: Yes

Reviewer #3: Yes

5. Is the manuscript presented in an intelligible fashion and written in standard English?

Reviewer #1: Yes

Reviewer #3: Yes

6. Review Comments to the Author

Reviewer #1: Title: Seasonality of Mansonia Blanchard (Diptera, Culicidae, Mansoniini) in Porto

Velho, State of Rondônia, Brazil, is relevant and is in line with the scope of the Journal. However, the following modifications need to be addressed to make it publishable: Manuscript Number: PONE-D-23-44055

The authors did not respond to the following question.

What was the criterion used to decide on the distances between the collection areas?

The authors emphasize the importance of selecting a study area for comprehensive point sampling.

Reviewer #3: I found the revised paper entitled 'Trends of Mansonia (Diptera, Culicidae, Mansoniini) in Porto Velho: seasonal patterns

and meteorological influences' to be of high quality and significance to entomology research. The clarity of presentation, the soundness of methods, and the novelty of findings should all be mentioned.

7. PLOS authors have the option to publish the peer review history of their article (what does this mean?). If published, this will include your full peer review and any attached files.

Reviewer #1: No

Reviewer #3: No

---

## [Author Response · Author response to Decision Letter 1]

7 Apr 2024

Reviewer #1: 

What was the criterion used to decide on the distances between the collection areas?

The authors emphasize the importance of selecting a study area for comprehensive point sampling.

Answer: 

The SkeeterVac traps were strategically installed based on three main considerations: 1. the presence of human communities, 2. the prevalence of Mansonia mosquito attacks, and 3. the presence of potential breeding sites, such as stagnant water bodies with aquatic macrophytes. We used satellite images to identify and georeferenced the lakes and lagoons close to the communities with the most of Mansonia mosquitoes attacks. The breeding sites were further validated by on-site surveys. This facilitated the determination of distances between traps and the nearest breeding sites.

Following installation based on these criteria, the spacing between traps was measured. A minimum spacing of 5km was maintained between traps, as exemplified by P08-P07-P06, while the greatest distances recorded were between points P05 and P06 (32km) and between points P01 and P02 (37) km. This spacing ensured no overlap in sampling, considering that the reported flight radius for Mansonia is on average 2km (de Mello et al., 2021).

Reference

de Mello, C.F., Alencar, J. Dispersion pattern of Mansonia in the surroundings of the Amazon Jirau Hydroelectric Power Plant. Sci Rep 11, 24273 (2021). https://doi.org/10.1038/s41598-021-03682-1

---

## [Editor Report · Decision Letter 2]

16 Apr 2024

Trends of Mansonia (Diptera, Culicidae, Mansoniini) in Porto Velho: seasonal patterns and meteorological influences

PONE-D-23-44055R2

Dear Dr. Saraiva,

We’re pleased to inform you that your manuscript has been judged scientifically suitable for publication and will be formally accepted for publication once it meets all outstanding technical requirements.

Kind regards,

Luca Nelli, PhD

Academic Editor

PLOS ONE